# Transcriptome Analyses Reveal the Key Regulators of Tomato Compound Leaf Development

**Guangwu Zhu, Chongtian Ma, Shuimiao Yu, Xueying Zhang, Jing Jiang**  **and Xin Liu \***

College of Horticulture, Shenyang Agricultural University, Shenyang 110866, China
\* Correspondence: liuxin890205@163.com; Fax: +86-24-88487166

**Abstract:** Leaves are one of the organs involved in plant assimilation and transpiration. Different leaf development processes can result in different leaf shapes. Tomato plants have typical compound leaves. It is helpful to explore the regulatory factors affecting the leaf development and morphogenesis of tomatoes to cultivate varieties with high photosynthetic efficiency. We used the typical tomato leaf shape mutants Petroselinum (*Pts*), Trifoliate (*tf2*), and Entire (*e*), which showed a gradual decrease in leaflet number and compound leaf complexity. Transcriptome sequencing was performed to analyze the key differentially expressed genes (DEGs) among the 3 groups, which revealed 2393, 1366, and 1147 DEGs in *Pts*/VF36, *tf2*/CR, and *e*/AC, respectively. We found 86 overlapping DEGs among the 3 groups. In addition, we found that the mutation of *Pts*, *tf2*, and *e* affected not only leaf morphology but also the wax, fatty acid, and abscisic acid pathways during growth and development. An RT-qPCR analysis during leaf primordium development revealed three transcription factors (*bHLH079*, *WRKY44*, and *WRKY76*) and three hormone-regulated genes (*IAA-amino acid hydrolase*, *Gibberellin2ox7*, and *Gibberellin20ox*) that were differentially expressed in the transcriptome. Using virus-induced gene silencing (VIGS), we observed the leaf shape of VIGS plants and found that *bHLH079*, *IAA-amino acid hydrolase*, *Gibberellin2ox7*, *Gibberellin20ox*, *WRKY44*, and *WRKY76* were the endogenous regulators influencing tomato compound leaf development. This study provides a promising direction for revealing the molecular regulation mechanism underlying compound leaf development in tomatoes.

**Keywords:** leaf shape; tomato; transcription factor; hormone



## 1. Introduction

Leaves are important aboveground plant parts and are the main sites for plant life activities. The leaf primordium initiates from the flanks of the shoot apical meristem (SAM) and develops into flat structures of variable sizes and forms. Leaf development is a continuous, dynamic, and complex process that can be divided into three stages: initial, primary morphogenesis (PM), and secondary morphogenesis (SM). During the initial stage, the leaf primordia emerge laterally from the SAM followed by transverse leaf expansion. During primary morphogenesis, the leaf spreads laterally; the basic leaf shape is determined; the transverse structure is refined from the leaf margin; and the leaf clefts, serrations, and leaflets are produced simultaneously [1]. During secondary morphogenesis, tissue differentiation and cell expansion occur, leading to increased leaf area eventually followed by a series of cell divisions and differentiation into mature leaves. Another important part of leaf morphogenesis is the establishment of leaf polarity during which cells continuously differentiate to eventually form asymmetric leaf organs [2].

Common tomato leaves are typical pinnate compound leaves, also known as uneven pinnate compound leaves, with deep or shallow lobes of irregular serration, and the size of the leaflets alternates. Furthermore, tomato leaves are of different types, such as entire, trifoliate, and compound leaves. Additionally, tomato leaves can be classified into entire, serrated, or notched leaves depending on their degree of serration [3,4]. The leaf margins in plants are also the result of long-term evolution. As an indispensable part of leaves, the

leaf margin is also very important to their morphology. The leaf margin of notched leaves is serrated. This morphological change can improve the stress resistance of plants under abiotic stress [5,6]. In addition, notched leaves have a larger specific leaf area than e leaves, so they are more competitive in their use of light resources [7].

During leaf growth, the developmental time course and morphogenesis are the result of both internal and external environmental factors, and each stage of leaf development is regulated by environmental factors, transcription factors, miRNAs, and plant hormones.

Several key transcription factors involved in the regulation of leaf organ development have been identified. For example, WOX family transcription factors belong to the homeodomain (HD) protein family and play important roles in embryogenesis and lateral organ development [8]. *SlLAM1* belongs to the WOX gene family, and *knockout SlLAM1*, which is obtained through CRISPR-Cas9-mediated genome editing, results in narrower tomato leaves and a reduction in the number of secondary lobules. Furthermore, the overexpression of SlLAM1 in tobacco partially restores the narrow leaf phenotype [9]. The KNOX (Knotted1-like homeobox) transcription factor family is a subfamily of the TALE protein family of homeobox genes which is involved in the regulation of plant growth and development as well as the formation and development of the SAM, and it is essential for the maintenance of SAM function [10]. KNOXI expression in compound leaf tomatoes inhibits differentiation during leaf development [11,12]; however, this inhibition delays leaf maturation and leads to the formation of more leaflets [13]. It is similar to the CIN-TCP family of transcription factors, which affects leaf morphology by regulating the leaf maturation time. The downregulation of the activity of TCP transcription factor *LANCEOLATE* (*LA*, *TCP2*) is essential for maintaining the genetic potential of leaf edge morphogenesis in tomatoes [14]. The LA gene is negatively regulated by miRNA319 at the early stage of leaf development, and when the sensitivity of LA mRNAs to miRNA319 recognition is reduced in the semidominant mutant, an increase in the LA expression level will promote leaf differentiation and lead to simpler leaves [15]. Two bHLH/HLH homolog proteins, leaf-associated protein 1 and leaf-associated protein 2 (*AtLP1* and *AtLP2*), have been identified in *Arabidopsis thaliana*; LP1 and LP2 play similar active roles in longitudinal cell elongation [16].

Plant hormones are key factors that regulate leaf shape. Auxin coordinates the phyllotaxis of leaf initiation from the SAM and determines the location of serrations and the initiation of the leaflets and lobes from the margin of the leaf primordia. Additionally, auxin influences leaf symmetry in both adaxial–abaxial [17] and bilateral [18] patterning. The ENTIRE (*SlIAA9*) gene in tomatoes, as a repressor of the auxin response, belongs to the Aux/IAA family of proteins [19]. In tomatoes, the Aux/IAA protein ENTIRE-regulated auxin response factors (ARFs) repress the auxin response and blade growth in the intercalary region. Loss-of-function *e* mutants develop single blades with partially fused primary leaflets due to ectopic blade growth [20–23].

During leaf development, GA regulates cell proliferation and expansion and leaf complexity. Increasing the GA level or GA reaction in tomatoes leads to premature leaf ripening and the formation of simpler leaves with smooth edges [24–28].

Cytokinins (CKs) are important regulators of development. During leaf development, CKs promote morphogenesis and delay cell differentiation, and CKs are also crucial for maintaining SAM function [29,30]. Conversely, reduced CK levels due to the expression of the CK degradation gene, *Cytokinin oxidase* (*CKX*), results in reduced leaf complexity. Genetic and molecular analyses have shown that CKs delay maturation downstream of the *KNOXI* transcription factor and mediate the activity of the KNOXI protein in leaf shape regulation [13,31,32]. The overexpression of the CK biosynthesis gene, *ISO-PENTENYLTRANSFERASE 7* (*IPT7*), in tomato leaves results in the formation of highly compound leaves [33].

Brassinosteroids (BRs) affect several developmental processes by promoting elongation and differentiation [34,35]. In tomatoes, the recessive BR-defective mutant dumpy (*dpy*) is short, causing dense, dark-green, furrowed leaves that curl downward [36]. There are few reports on the involvement of jasmonic acid (JA) [37], abscisic acid (ABA) [38], ethylene (ET) [39,40], and other hormones in tomato leaf development.

In this study, VF36, Petroselinum (*Pts*), Condine Red (CR), Trifoliate (*tf2*), Ailsa Craig (AC), and Entire (*e*) were used as experimental materials. Due to the gradual decrease in the leaflet number and compound leaf complexity in the three groups of mutants, the key differentially expressed genes (DEGs) were screened via a transcriptome sequencing analysis. The transcriptome analysis revealed 86 overlapping DEGs among the 3 groups. It was found that the mutations of *Pts*, *tf2*, and *e* were not only reflected in leaf morphology but also affected the wax, fatty acid, and abscisic acid pathways during plant growth and development. Moreover, the DEGs from the *Pts*/VF36-*tf2*/CR, *Pts*/VF36-*entire*/AC, and *tf2*/CR-*entire*/AC groups were selected for further RT-qPCR analysis. The RT-qPCR results during leaf primordium development showed that the six key factors had a significant expression trend. Finally, the main regulators of tomato leaf shape were selected by observing plants with virus-induced gene silencing (VIGS).

## 2. Results

### 2.1. Leaf Shape Characteristics of the Mutant Plants

The tomato mutant plants (*Petroselinum*, *trifoliate*, and *entire*) showed differences in the number of their compound leaves and in their leaf margins compared to the wild types. Compared with wild-type VF36, the *Pts* mutant had a significantly greater number of lobules on the leaf axis (Figures 1A and S1A). In contrast, the *tf2* mutant showed a simple three-leaflet state with a single narrow leaf in the early stage and a pair of lateral leaves on the petiole in the late stage. Additionally, no leaflets were present (Figures 1B and S1B). Compared with AC, which had a typical compound leaf, the *e* mutant showed functional defects, leading to the development of single leaves with smooth leaf margins and significantly greater blade lengths and widths (Figures 1C and S1C). The three mutants showed that the complexities of *Pts*, *tf2*, and *e* ranged from high to low. Therefore, these mutants were used to study the relationship between leaf complexity and gene expression.

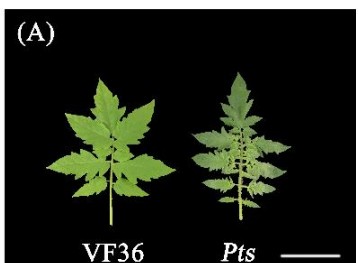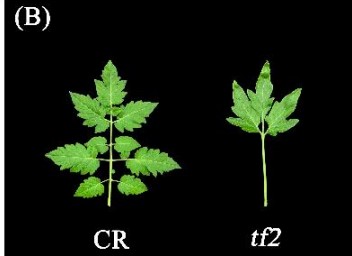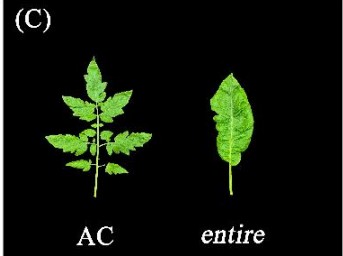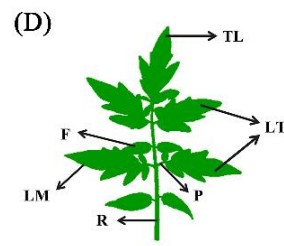

**Figure 1.** Morphological differences between mutant and wild-type plants. (**A**) *Pts* mutant had significantly greater number of leaflets than VF36 wild type. (**B**) *tf2* mutant had significantly smaller number of leaflets and deeper cleft than CR wild type. (**C**) *entire* mutant had simplified leaves and smooth leaf margins compared with AC wild type. (**A–C**) Plants were 45 d old. The selected leaf was the fifth true leaf of the 45 d old plants in (**A–C**). Bars = 500 μm in (**A–C**). (**D**) Leaf structure. TL represents terminal leaflet, LT represents lateral leaflet, F represents foliole, P represents petiolated, R represents rachis, and LM represents leaf margin.

### 2.2. The Simplification and Complexity of Leaves Can Affect Leaf Hair Development and Cell Division

The delay or promotion of leaf development affected leaf complexity. An increase in the number of compound leaves in the *Pts* mutant enhanced leaf complexity, whereas the *tf2* and *e* mutants showed reduced leaf complexity due to a reduction in the number

of compound leaves. Compared with tomatoes, potatoes, eggplants, and peppers, the premature development of the trichomes or the premature erect leaf primordia formed relatively simple leaves [15]. Scanning electron microscopy showed that the leaf hair number of *Pts* was significantly lower than that of the wild type, whereas the number of leaf hairs in *tf2* and *e* was significantly higher than that in the wild types (Figure 2A–G). We analyzed the leaf cells to investigate whether premature or delayed leaf maturation affected the leaf epidermal cell levels. The results showed that the number of epidermal cells in the *Pts* mutant plant was significantly higher than that in the wild-type plant, but the cell size was smaller. The number of epidermal cells in the *tf2* and *e* mutants decreased significantly, and the cells were larger (Figure 2H–T). These results indicate that early leaf maturation simplifies leaves, increases leaf hairs, decreases the division ability of epidermal cells, and increases the expansion of epidermal cells, whereas delayed leaf maturation decreases leaf hairs, increases the division ability of epidermal cells, prolongs the division period, and shortens the expansion period.

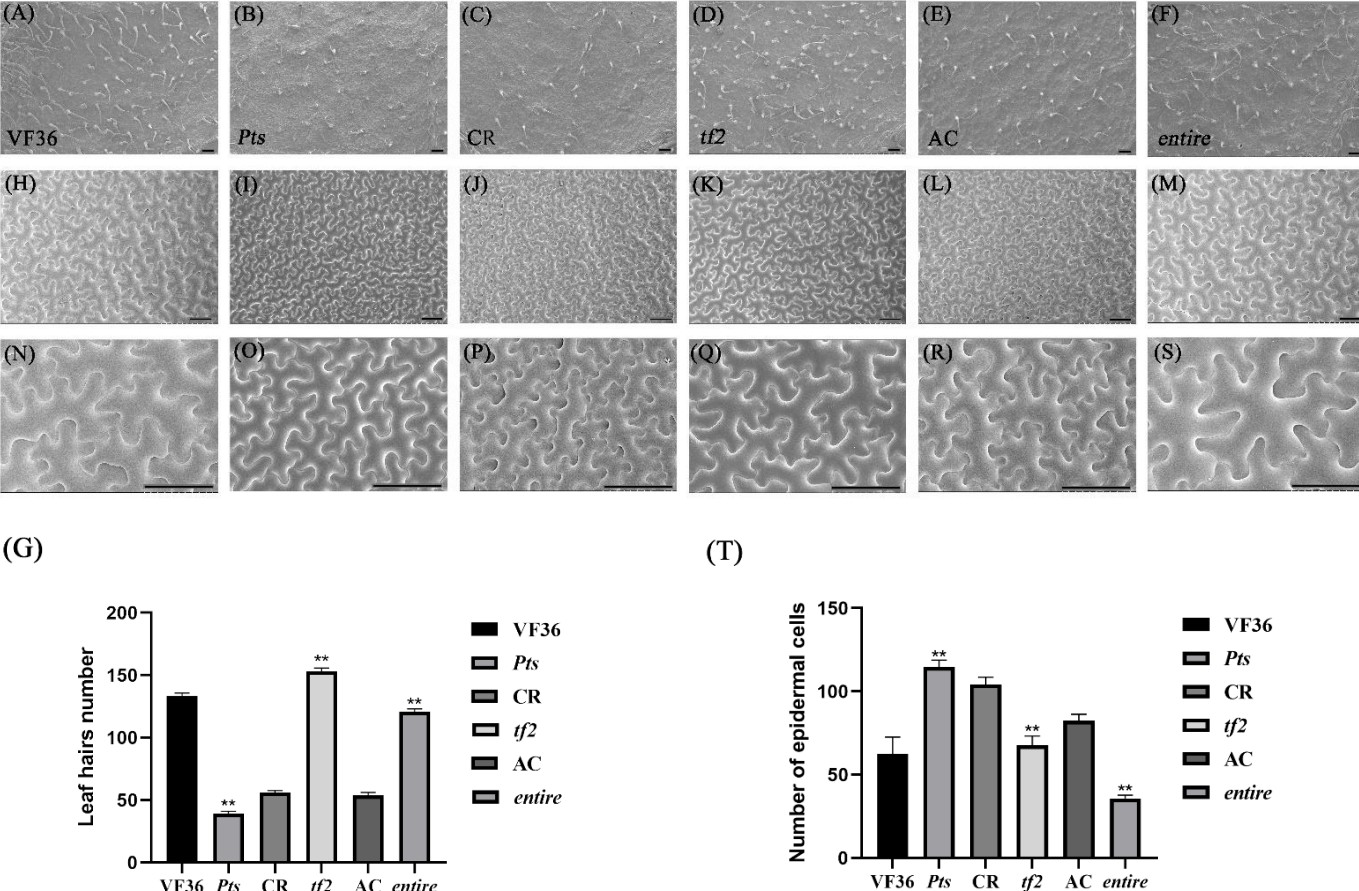

**Figure 2.** Differences in the number of leaf hairs and cells between tomato mutants and wild types visualized using scanning electron microscope (SEM). (**A**–**F**) Images of mutant *Pts*, *tf2*, and *e* and wild-type VF36, CR, and AC under SEM. (**G**) Differences in the number of leaf hairs between mutant *Pts*, *tf2*, and *e* and wild-type VF36, CR, and AC. (**H**–**S**) Images of epidermal cells observed under SEM at 300 (**H,J,L,N,P,R**) and 800 (**I,K,M,O,Q,S**) magnification. (**T**) The number of epidermal cells differed between mutant *Pts*, *tf2*, and *entire* and wild-type VF36, CR, and AC. (**G,T**) Values are presented as means $\pm$ SE ($n = 3$). ** indicate significant differences between wild types and transgenic lines or mutants at $p < 0.05$ and $p < 0.01$, respectively. (**A**–**S**) Bars = 50 µm.

### 2.3. Transcriptome Sequencing among Pts/VF36, tf2/CR, and e/AC Groups

To identify the DEGs in the tomato mutants with different leaf shapes, we took the fifth true leaf at 45 days. A total of 18 RNA libraries were constructed from *Pts*, *tf2*, *e*, and their

respective wild-type leaf samples. Raw reads were generated via the high-throughput sequencing of the six samples with three replicates each (Table S1). After data filtering, the useful reads of the sequencing results for each cDNA library accounted for >87.03% of the raw reads (Q30 > 91.78%) (Table S1). Among the 18 samples, 94.63–96.08% of the reads could be mapped to the tomato genome 4.0 (https://solgenomics.net, on 2 December 2022). Of these mapped reads, 93.84–97.23% matched uniquely. The transcriptome analysis showed 86 overlapping DEGs in the 3 groups—*Pts*/VF36, *tf2*/CR, and *e*/AC; 250 DEGs were detected in the *Pts*/VF36 and *tf2*/CR groups, 257 were detected in the *Pts*/VF36 and *entire*/AC groups, and 147 were detected in the *tf2*/CR and *entire*/AC groups (Figure 3A). A systematic analysis, a database comparison, gene annotation, data acquisition, and a prediction were performed on the 86 genes. Finally, eight candidate genes were screened according to the trend of increasing or decreasing differences in gene expression among the three groups—*Pts*/VF36, *tf2*/CR, and *e*/AC (Table 1). We found four GDSL genes, one MADS transcription factor family gene, one abscisic acid receptor (PYL4) gene, and two fatty-acid-related genes.

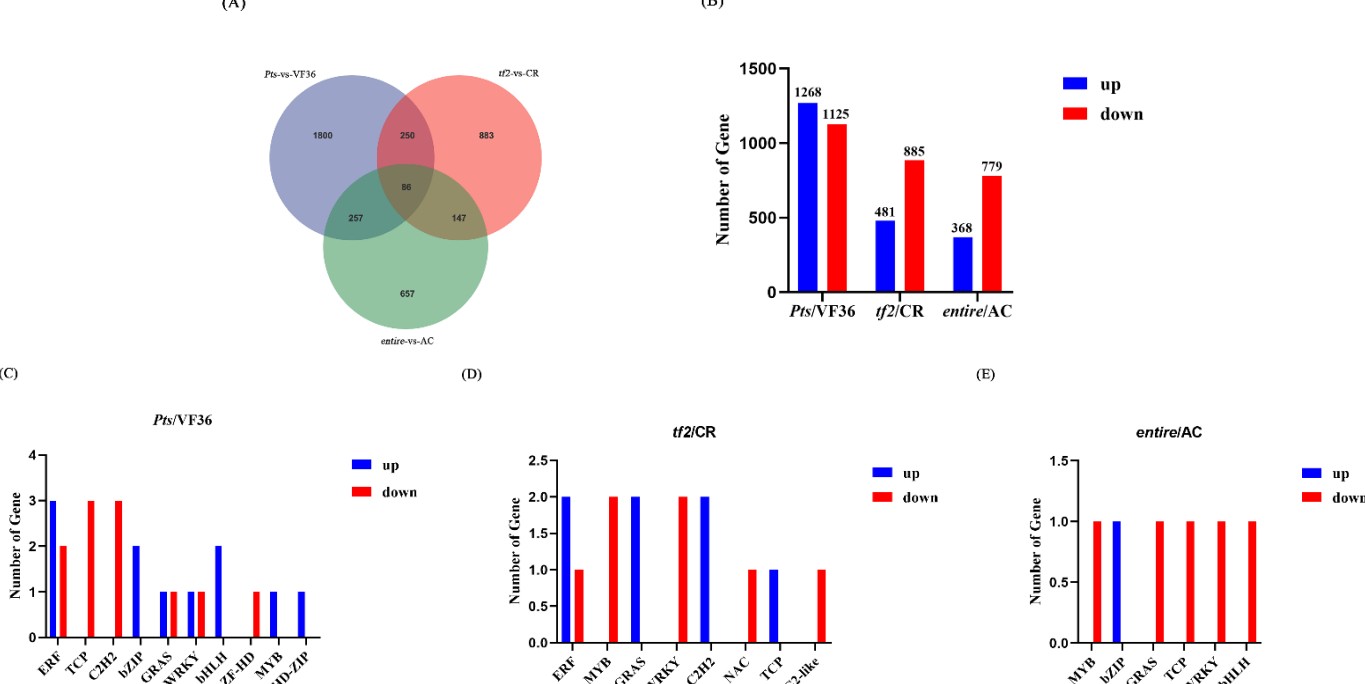

**Figure 3.** Overlapping differential genes and transcription factor families of differentially expressed genes (DEGs) in *Pts*/VF36, *tf2*/CR, and *e*/AC. (**A**) Number of DEGs in transcriptome of tomato mutant plants with different leaf morphology compared with that of the wild-type plants. The sum of the numbers in each circle is the total number of DEGs in the comparison combination, and the overlapping parts of the circles represent the DEGs shared between the two comparison groups. (**B**) The abscissa represents the comparison groups of difference analysis, the ordinate represents the number of DEGs, blue bars represent upregulated expression, and red bars represent downregulated expression. (**C**–**E**) The transcription factor families were differentially expressed in *Pts*/VF36, *tf2*/CR, and *e*/AC. The abscissa represents different transcription factor families, and the ordinate represents the number of different genes falling into a transcription factor family. Blue bars represent upregulated expression, and red bars represent downregulated expression.

**Table 1.** RNA-Seq analysis of differentially expressed genes (DEGs) in *Pts*/VF36, *tf2*/CR, and *entire*/AC.

| Gene ID | *Pts*/VF36 | Log$_2$ (FC) | *tf2*/CR | Log$_2$ (FC) | *entire*/AC | Log$_2$ (FC) | Description |
|---|---|---|---|---|---|---|---|
| Solyc06g064820.3 | Downregulation | −2.30 | Downregulation | −3.52 | Downregulation | −5.58 | GDSL esterase/lipase *At1g71691* |
| Solyc11g031960.2 | Downregulation | −1.36 | Downregulation | −1.79 | Downregulation | −1.80 | GDSL esterase/lipase *At1g33811* |
| Solyc11g006250.2 | Downregulation | −1.71 | Downregulation | −3.50 | Downregulation | −3.63 | GDSL esterase/lipase *At5g33370* |
| Solyc01g095450.4 | Downregulation | −1.94 | Downregulation | −3.42 | Downregulation | −5.72 | GDSL esterase/lipase *At4g01130* |
| Solyc03g006830.3 | Downregulation | −1.89 | Downregulation | −2.37 | Downregulation | −2.50 | MADS-box transcription factor 2 |
| Solyc05g009270.4 | Downregulation | −3.40 | Downregulation | −3.84 | Downregulation | −4.03 | Fatty acid elongase 3-ketoacyl-CoA synthase |
| Solyc09g090500.3 | Downregulation | −1.42 | Downregulation | −1.91 | Downregulation | −2.10 | Cyclopropane-fatty-acyl-phospholipid synthase |
| Solyc06g050500.2 | Downregulation | −1.26 | Upregulation | 1.66 | Upregulation | 1.82 | Abscisic acid receptor PYL4 |

The transcriptome analysis showed 2393 DEGs in *Pts*/VF36, 1268 upregulated genes and 1125 downregulated genes; 1366 DEGs in *tf2*/CR, 481 upregulated genes and 885 downregulated genes; and 1147 DEGs in *entire*/AC, 368 upregulated genes and 779 downregulated genes (Figure 3B). The transcription factor families, such as WOX, TCP, bHLH, MADS, WRKY, and MYB, were identified via the analysis of the differentially expressed transcription factors (Figure 3C–E). A total of seven DEGs (bHLH155, MYB13, MADS, MADS-2, TCP12, TCP13, and TCP22) were screened from these transcription factor families (Tables S3 and S4). As described in the Introduction, these transcription factor families are involved in leaf development.

We performed Gene Ontology (GO) and Kyoto Encyclopedia of Genes and Genomes (KEGG) enrichment analyses to further analyze the potential biological roles of the DEGs. We found that the DEGs in *Pts*/VF36 were mainly enriched in the cell periphery/GO:0071944, the plasma membrane/GO:0005886, transcription regulator activity/GO:0140110, DNA-binding transcription factor activity/GO:0003700, the small-molecule biosynthetic process/GO:0044283, the fatty acid biosynthetic process/GO:0006633, and the cell/GO:0005623 (Figure S2A). The DEGs in *tf2*/CR were mainly enriched in catalytic activity, oxidoreductase activity/GO:0016491, and the oxidation–reduction process/GO:0055114 (Figure S2B). The DEGs in *entire*/AC were mainly enriched in the thylakoid/GO:0009579, the chloroplast thylakoid/GO:0009534, the thylakoid membrane/GO:0042651, the photosystem/GO:0009521, and photosynthesis/GO:0015979 (Figure S2C). In the KEGG enrichment analysis, we found that the DEGs in *tf2*/CR were mainly enriched in amino sugar and nucleotide sugar metabolism, phenylpropanoid biosynthesis, and the MAPK signaling pathways, whereas the DEGs in *Pts*/VF36 and *entire*/AC were mainly enriched in plant hormone signal transduction (Figure S2D–F).

### 2.4. Transpiration Rate, Stomatal Conductance, and Epidermal Cell Permeability Are Closely Related to GDSL Esterase/Lipase Expression

As mentioned above, the four GDSL genes showed a gradual downregulation trend in the three transcriptomes (Table 1). Previous studies have shown that GDSL genes are related to wax biosynthesis and stomatal outer cuticular ledge (OCL) formation [41]. Therefore, we analyzed the corresponding indices of the stomata. The number of stomata in *Pts* and *e* was significantly higher than that in the wild types, with no significant difference in the number of stomata between *tf2* and the wild type (Figure 4B). However, the transpiration rate and stomatal conductance of the three mutants were significantly decreased compared to those of the wild types (Figure 4A,C). Based on previous studies, we divided the stomata into three types: type I, normally opened stomatal pores; type II, stomatal pores that were not fully occluded; and type III, stomatal pores that were fully covered [41]. We found that the stoma status of the mutants and their wild types differed according to the statistics of the three types of stomata (Figure 4G). Although the number of stomata changed significantly, the opening and closing state of the stomata was the key factor affecting the transpiration rate. The chlorophyll leaching rate was positively correlated with cell permeability. The results of the chlorophyll leaching experiment further verified that the cell permeability of the mutants was enhanced. The leaching rate of the *Pts* mutant was significantly higher than that of the wild type at 10–90 min (Figure 4D). The leaching rate of *tf2* was significantly higher than that of the wild type at 10–30 min, and the subsequent time was similar to that of the wild type (Figure 4E). The leaching rate of mutant *e* was significantly higher than that of the wild type at 10 min, 30 min, and 50 min but was significantly lower than that of the wild type at 70–90 min (Figure 4F). We measured the total chlorophyll content at 24 h and found no significant difference in the *Pts* and *tf2* mutants compared with their respective wild types, but the total chlorophyll content of mutant *e* was significantly lower than that of the wild type (Figure 4H). Thus, the leaching rate of mutant *e* at 70–90 min was significantly lower than that of the wild type.

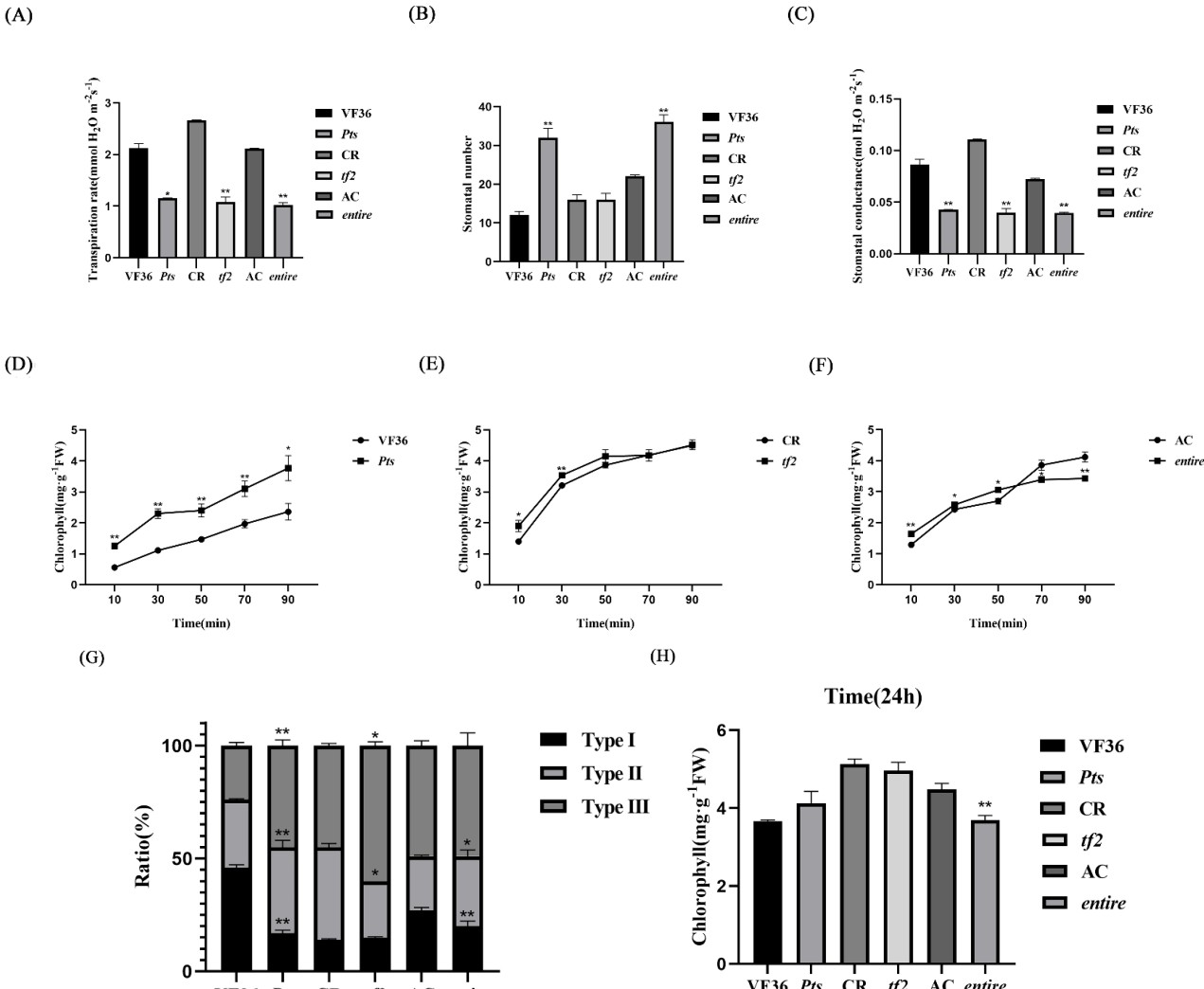

**Figure 4.** Physiological differences between the mutants and the wild types. (**A**) Transpiration rate of the fourth true leaf. Mutants *Pts*, *tf2*, and *e* and their corresponding wild types VF36, CR, and AC. Values are presented as means ± SE (*n* = 3). * and ** indicate significant differences between wild types and transgenic lines or mutants at *p* < 0.05 and *p* < 0.01, respectively. (**B**) Differences in stoma number between mutants *Pts*, *tf2*, and *e* and their corresponding wild types VF36, CR, and AC. Values are presented as means ± SE (*n* = 3). * and ** indicate significant differences between wild types and transgenic lines or mutants at *p* < 0.05 and *p* < 0.01, respectively. (**C**) Differences in stomatal conductance between mutants *Pts*, *tf2*, and *entire* and their corresponding wild types VF36, CR, and AC. Values are presented as means ± SE (*n* = 3). * and ** indicate significant differences between wild types and transgenic lines or mutants at *p* < 0.05 and *p* < 0.01, respectively. (**D–F**) Determination of chlorophyll leaching rate. Measurements were recorded every 20 min in the mutants *Pts*, *tf2*, and *e* and their corresponding wild types VF36, CR, and AC. Values are presented as means ± SE (*n* = 3). * and ** indicate significant differences between wild types and transgenic lines or mutants at *p* < 0.05 and *p* < 0.01, respectively. (**G**) The ratio of type I, type II, and type III stomata in the mutants *Pts*, *tf2*, and *e* and their corresponding wild types VF36, CR, and AC. type I, normal opened stomatal pores; type II, stomatal pores not fully occluded; type III, stomatal pores fully covered. Values are presented as means ± SE (*n* = 3). * and ** indicate significant differences between wild types and transgenic lines or mutants at *p* < 0.05 and *p* < 0.01, respectively. (**H**) Total chlorophyll content at 24 h in mutants *Pts*, *tf2*, and *e* and their corresponding wild types VF36, CR, and AC. Values are presented as means ± SE (*n* = 3). * and ** indicate significant differences between wild types and transgenic lines or mutants at *p* < 0.05 and *p* < 0.01, respectively.

*2.5. Differentially Expressed Genes in Pts/VF36 and tf2/CR, Pts/VF36 and e/AC, and tf2/CR and e/AC*

The above results indicate that GDSL was not the key factor affecting leaf morphology but that the expression of GDSL could be changed due to different leaf morphologies. Therefore, to further determine the key factors affecting leaf development, we compared the DEGs in groups of two of the transcriptomes to expand the scope of screening. By analyzing the overlapping DEGs, we found five DEGs in the *Pts*/VF36 and *tf*2/CR comparison group (Table S6) and two in the *Pts*/VF36 and *entire*/AC comparison group (Table S7), which was similar to that in the *tf*2/CR and *entire*/AC comparison group (Table S8). The DEGs were identified as bHLH079, IAA-aah, GA2ox7, GA20ox, WRKY44, and WRKY76. Auxin and gibberellic acid (GA) play key roles in leaf development. Auxin coordinates the phyllotaxis of leaf initiation from the SAM and determines the location of serrations and the initiation of the leaflets and lobes from the margin of the leaf primordia [42]. In leaf development, GA regulates cell proliferation and expansion and leaf complexity [24,25,27,43]. The transcription factors of the bHLH and WRKY families have also been shown to play roles in leaf development. *AtLP1* and *AtLP2*, two bHLH/HLH homologs in *Arabidopsis thaliana*, positively regulate leaf cell elongation [16], and *NtWRKY4* participates in leaf morphogenesis [44].

*2.6. IAA and GA May Regulate Compound Leaf Shape Formation in Tomatoes*

Plant hormones are key factors affecting leaf development and morphogenesis. As mentioned above, IAA-aah and GA2ox7 were differentially expressed in *Pts*/VF36 and *tf*2/CR, and GA20ox was differentially expressed in *Pts*/VF36 and *e*/AC. Therefore, we selected these two hormones for further analysis. The auxin and $GA_3$ content increased in *Pts*/VF36, and the auxin and $GA_3$ content decreased in *entire*/AC (Figure 5A,B). In the IAM pathway of auxin synthesis, the upregulation of AMIE gene expression in *Pts* led to an increased auxin content. The synthesis of the *tf2* and *e* proteins was decreased due to the downregulation of the expression of the AMIE gene (Figure 5C). In the GA biosynthesis pathway, the upregulation of GA2ox gene expression in *Pts* resulted in reduced GA decomposition ability and increased GA synthesis, whereas the opposite was observed for *e*. The upregulation of the expression of CYP701 in *tf2* increased $GA_{12}$ synthesis by GGDP, which is the precursor in GA synthesis, thereby increasing the content of GA (Figure 5D). To date, more than 130 GAs have been identified among which $GA_1$, $GA_3$, $GA_4$, and $GA_7$ have the highest bioactivity. Therefore, we estimated the content of $GA_3$ in the tomato mutant and wild-type plants. The $GA_3$ content in *Pts* and *tf2* was higher than that in the wild-type plants, whereas the total content was lower than that in the wild-type plants. To further verify this result, we analyzed the biosynthesis pathway of GA using KEGG and found two key gene groups, cytochrome P450 monooxygenases (CYP701) and GA2ox. In higher plants, GA synthesis is mainly divided into three stages. In plastids, ent-copalyl diphosphate synthase (CPS) and ent-kaurene synthase (KS) catalyze the conversion of the precursor GGPP to ent-kaurene. In the endoplasmic reticulum, the conversion of ent-kaurene to $GA_{12}$, which is the first $C_{20}$-GA in the biosynthetic pathway, is catalyzed by two cytochrome P450 monooxygenases, ent-kaurene oxidase (KO) and ent-kaurenoic acid oxidase (KAO) [45,46], and KO belongs to the subfamily CYP701A. In the cytoplasm, GA20ox and GA3ox eventually catalyze $GA_{12}$ into $GA_3$ with high biological activity (Figure 6D).

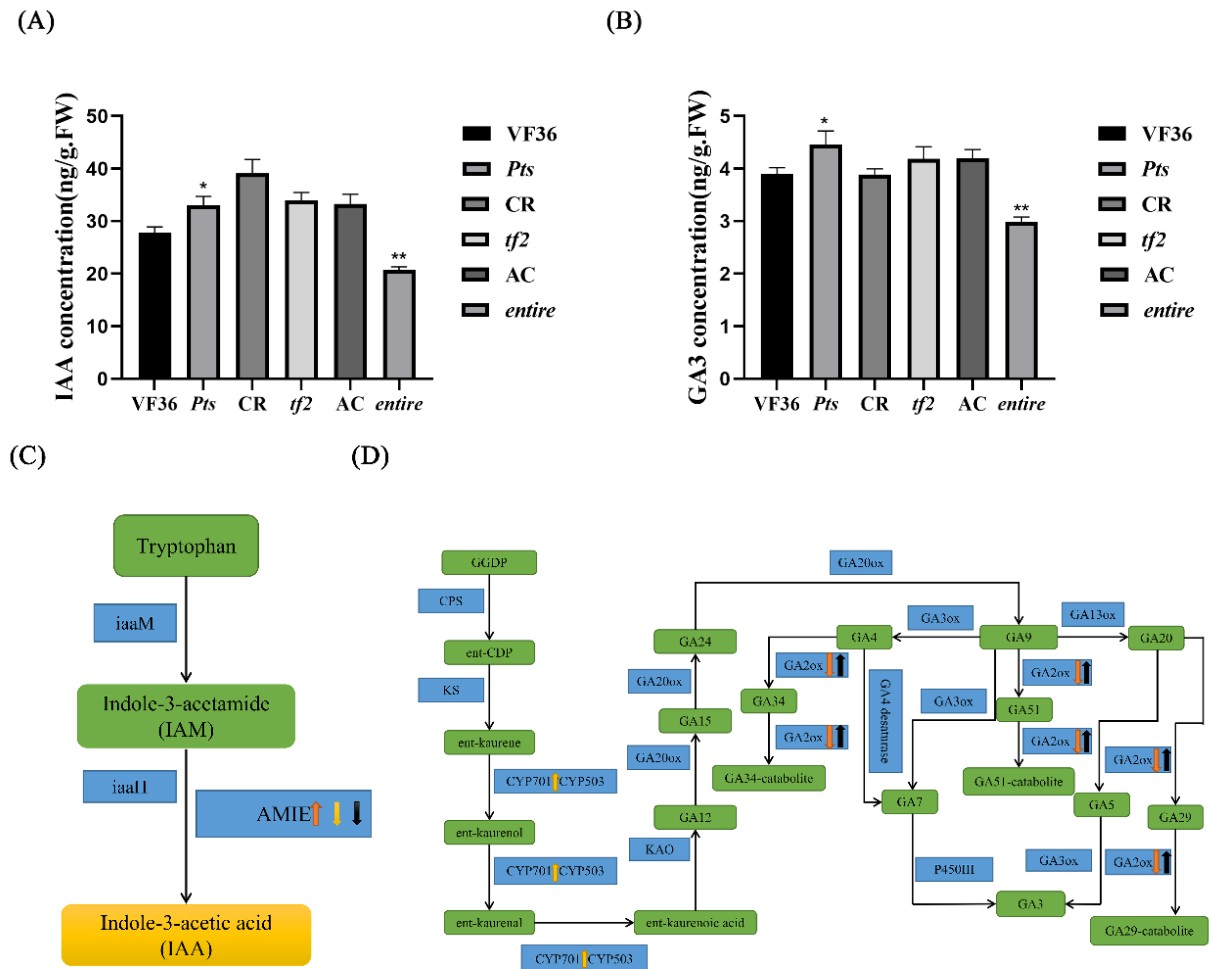

**Figure 5.** Determination of concentration of IAA and GA$_3$ and biosynthesis pathway analysis. (**A**,**B**) The contents of IAA and GA$_3$ in mutants *Pts*, *tf2*, and *e* and in their corresponding wild types VF36, CR, and AC were determined. Values are presented as means ± standard error (SE) (*n* = 3). * and ** indicate significant differences between wild types and transgenic lines or mutants at *p* < 0.05 and *p* < 0.01, respectively. (**C**,**D**) Related gene changes in IAA and GA$_3$ biosynthesis pathways. The orange arrows represent *Pts*/VF36, yellow arrows represent *tf2*/CR, and black arrows represent *e*/AC. Upward arrows represent upregulation of gene expression, and downward arrows represent downregulation of gene expression.

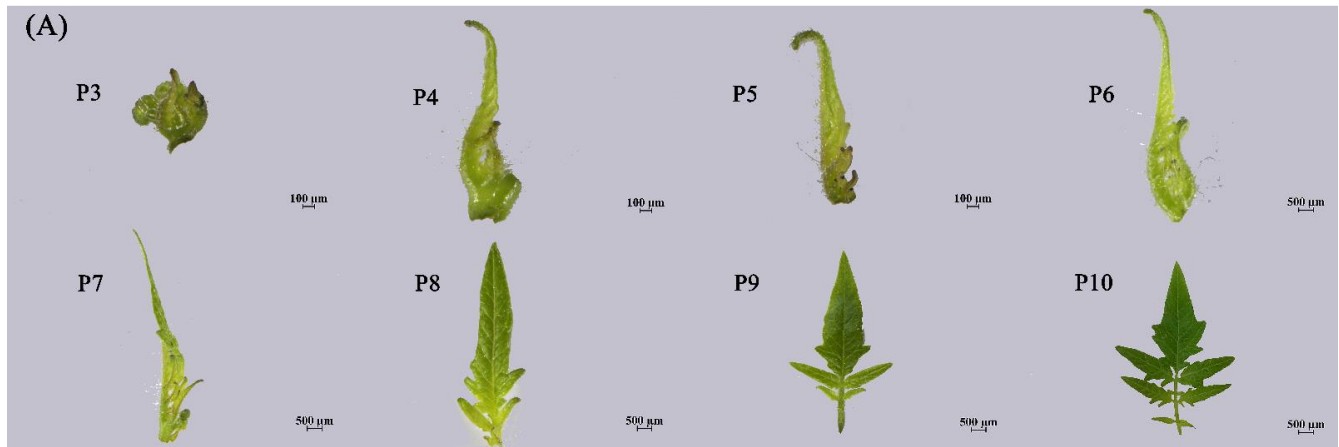

**Figure 6.** *Cont.*

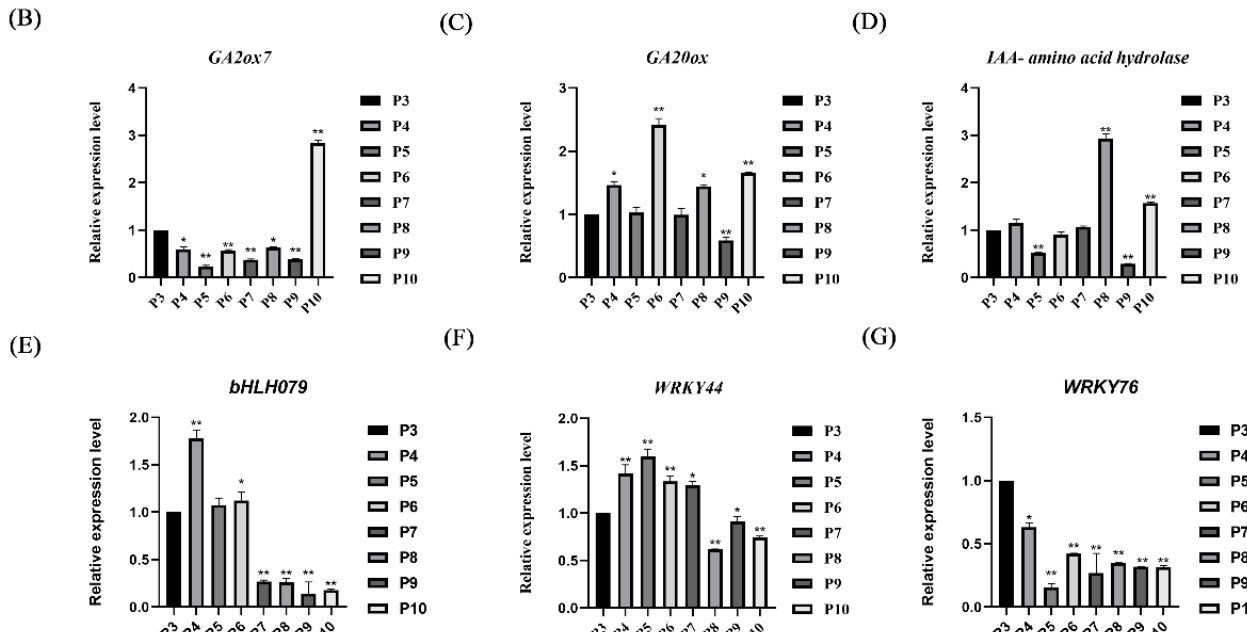

**Figure 6.** Analysis of the expression levels of candidate genes at different stages of AC leaf primordium development. (**A**) The P3–P10 stages of leaf primordium development were selected, and AC was used as the material. Bars = 500 μm in (**A**). (**B–G**) Analysis of the expression levels of candidate genes at different developmental stages of leaf primordia. The gene expression level at P3 was selected as the control. Mean values and standard deviation from three biological replicates are shown. Independent *t*-tests demonstrated a significant (* $p < 0.05$) or a highly significant (** $p < 0.01$) difference in gene expression levels.

## 2.7. Identification of the Expression of Candidate Genes at Different Leaf Development Stages in Tomatoes

As mentioned above, we screened six genes by comparing the transcriptomes in groups of two. We selected bHLH079, IAA-aah, GA2ox7, GA20ox, WRKY44, and WRKY76 as candidate genes. To verify whether these are involved in leaf development, we performed RT-qPCR to analyze their expression at the P3–P10 stages of leaf primordium development (Figure 6A–G). Six genes were found to be differentially expressed during leaf primordium development. Specifically, bHLH079 was highly expressed at the P4 stage and was significantly downregulated at the P7–P10 stages (Figure 6E). IAA-aah was highly expressed at the P8 stage, was significantly downregulated at the P9 stage, and was significantly upregulated at the P10 stage (Figure 6D). GA2ox7 expression was significantly downregulated at the P4–P9 stages and was significantly overexpressed at the P10 stage (Figure 6B). GA20ox expression was significantly upregulated at the P4, P6, P8, and P10 stages and was downregulated at the P9 stage (Figure 6C). WRKY44 expression was significantly upregulated at the P4–P7 stages and was downregulated at the P8–P10 stages (Figure 6F). WRKY76 expression was significantly downregulated at the P4–P10 stages (Figure 6G). Therefore, bHLH079, IAA-aah, GA2ox7, GA20ox, WRKY44, and WRKY76 may play important roles in the regulation of tomato compound leaf development. Next, these six genes were verified using VIGS.

## 2.8. Morphological Changes in Plants with VIGS

The plants with VIGS were identified using RT-qPCR (Figure 7A–F). Compared with the plants containing the empty vector pTRV2, we found that the *pTRV2-SlGA2ox7*-containing plants were taller, whereas the *pTRV2-SlGA20ox-*, *pTRV2-SlWRKY44-*, *pTRV2-SlWRKY76-*, *pTRV2-SlbHLH079-*, and *pTRV2-SlIAA-aah*-containing plants were shorter (Figure S3A–G). The *pTRV2-SlGA20ox*-containing plants were significantly dwarfed, with

shorter internodes, smaller leaves, and fewer secondary lateral lobules (Figure 8C). The *pTRV2-SlWRKY44-* and *pTRV2-SlWRKY76*-containing plants showed a dwarf phenotype, with smooth leaf margins, reduced serrations, leaf blade yellowing, a decreased number of leaflets, and no obvious petioles on the individual primary leaflets (Figure 8F,G). The *pTRV2-SlIAA-aah-* and *pTRV2-SlbHLH079*-containing plants were also dwarfed, with smaller leaves. Additionally, the *pTRV2-SlIAA-aah* leaves had deep fission (Figure 8D), whereas the *pTRV2-SlbHLH079* leaves had smooth blade edges and a reduced number of blades (Figure 8E).

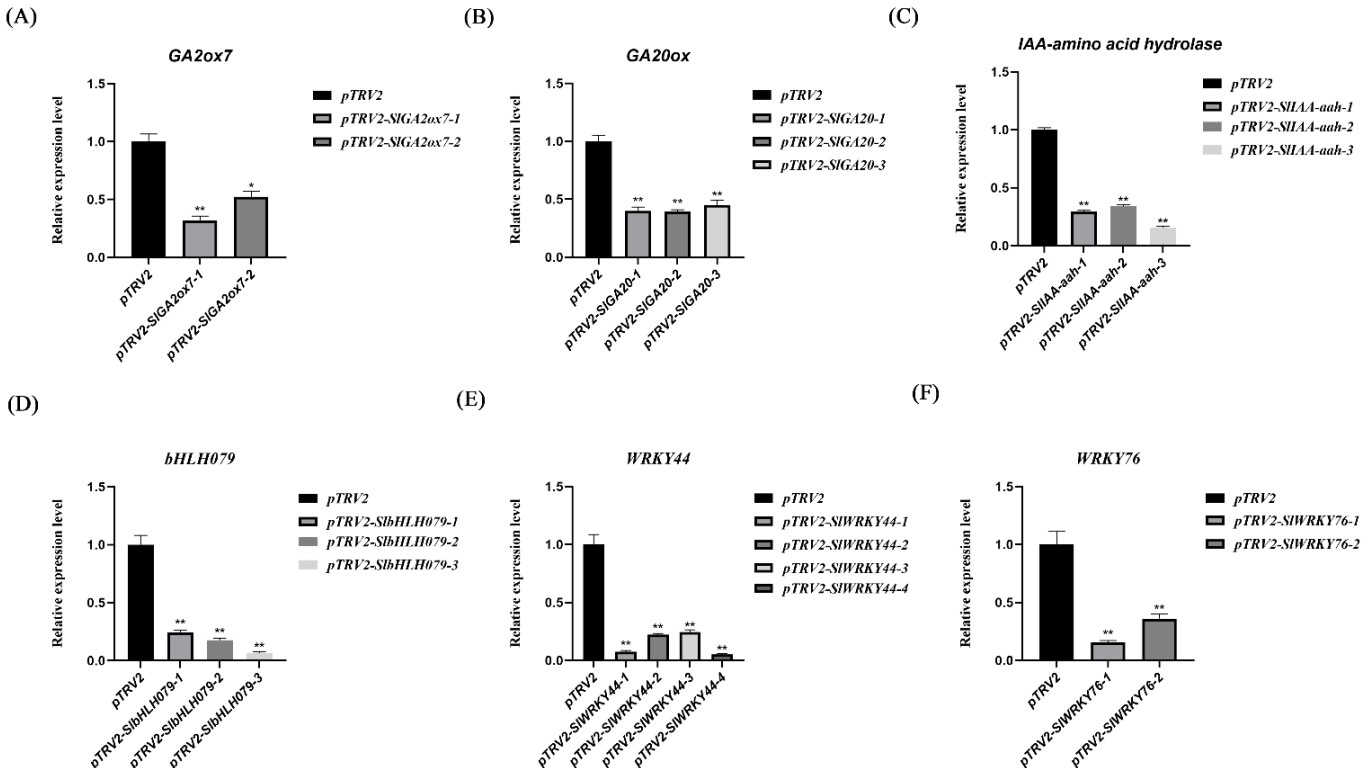

**Figure 7.** Gene expression in plants with VIGS compared with that in pTRV2 control. (**A**–**F**) The expression of *bHLH079*, *IAA-Amino acid hydrolase*, *Gibberellin2ox7*, *Gibberellin20ox*, *WRKY44*, and *WRKY76* in VIGS plants was compared with that in pTRV2 plants. pTRV2 was used as a blank vector control. Mean values and standard deviation from three biological replicates are shown. Independent *t*-tests demonstrated a significant (* $p < 0.05$) or a highly significant (** $p < 0.01$) difference in gene expression levels.

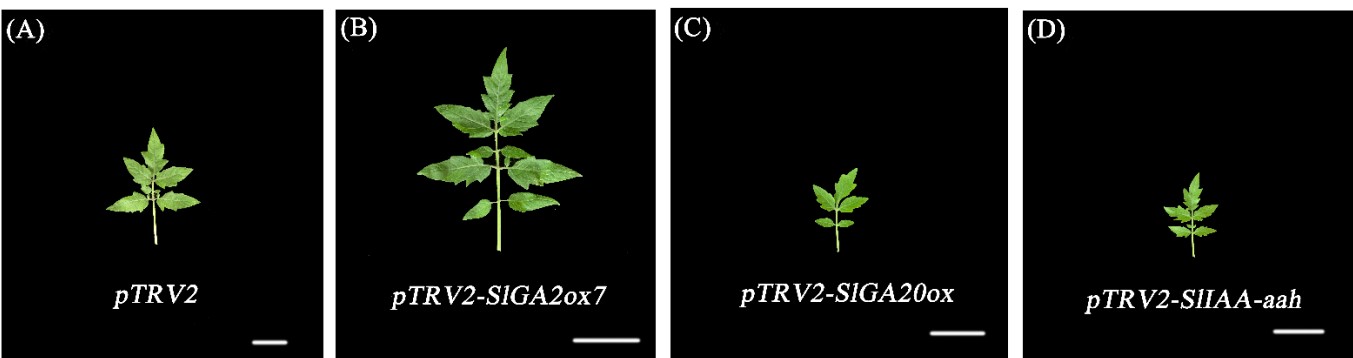

**Figure 8.** *Cont.*

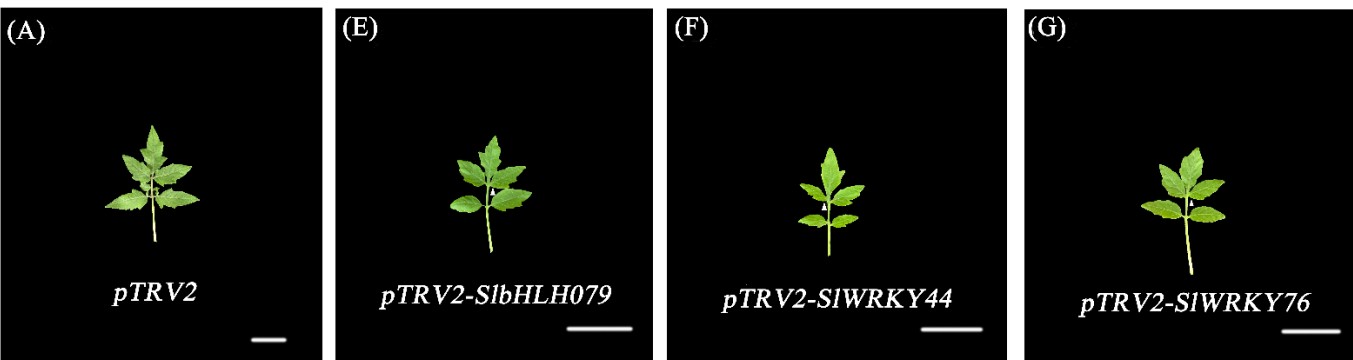

**Figure 8.** Differences in leaf morphology of VIGS plants compared with that of pTRV2 plants. (**A–G**) The fourth true leaf of VIGS and pTRV2 plants at 45 d were used as the control. Bars = 4 cm.

## 3. Discussion

### 3.1. Different Compound Leaf Shapes Influenced GDSL Expression in Tomatoes

To explore the key factors affecting leaf development, we selected three groups of tomato plants with different leaf morphologies for a transcriptome analysis, and four GDSL genes (*Solyc06g064820.3*, *Solyc11g031960.2*, *Solyc11g006250.2*, and *Solyc01g095450.4*) were identified through the analysis of three overlapping DEGs (Table 1). It is noteworthy that both the number of compound leaves and the complexity of the leaves of *Pts*, *tf2,* and *e* gradually decreased, and the expression of the four GDSL genes also showed the same trend in the three groups. GDSL esterases/lipases play a key role in regulating plant growth and development, stress resistance, and tissue and organ morphogenesis. Some studies have shown that the GDSL genes affect the differentiation of epidermal cells and are involved in wax biosynthesis and stomatal OCL formation. The stomatal cells and the pavement cells of the rice mutant Wilted dwarf and Lethal 1 (*WDL1*) are small [47], and Arabidopsis OSP1 (Occlusion of Stomata 1) mutants lead to decreased leaf wax synthesis, leading to stomatal occlusion, which leads to increased epidermal permeability, a decreased transpiration rate, and increased drought resistance [41]. The transpiration rate (Figure 4A), stomatal conductance (Figure 4C), and epidermal permeability (Figure 4D–F) of *Pts*, *tf2*, and *e* were significantly lower than those of the wild types, which is consistent with previous findings. Although the mutants differed from the wild types in the number of total stomata (Figure 4B), we believe that the different types of stomatal (Figure 4G,H) opening and closing states play a key role in stomatal conductance and the transpiration rate. These results indicate that GDSLs are closely related to the transpiration rate, stomatal conductance, and cell permeability. Although GDSLs are not the key factors directly affecting leaf morphology, differences in leaf morphology will affect the expression of the GDSL genes to further change the corresponding leaf physiological functions.

### 3.2. The Roles of the Plant Hormones IAA and GA in Tomato Compound Leaf Development

To further screen the key factors that may affect leaf morphology, we compared the transcriptomes in groups of two. For the comparison of the groups, we selected six genes—*IAA-aah*, *GA2ox7*, *GA20ox*, *bHLH079*, *WRKY44*, and *WRKY76*. The six genes can be divided into two groups—hormone-related genes and transcription factors. The expression of *IAA-aah* and *GA2ox7* decreased gradually in the *Pts*/VF36 and *tf2*/CR comparison group (Table S6), while *GA20ox* expression increased gradually in the *Pts*/VF36 and *e*/AC comparison group (Table S7). Auxin and GA play key roles in leaf development. Auxin coordinates the phyllotaxis of leaf initiation from the SAM and determines the location of serrations and the initiation of the leaflets and lobes from the margin of the leaf primordia [42,48,49]. The regulation of the auxin pathway during tomato leaf development can lead to a significantly simplified leaf phenotype. The exogenous application of 2-4-D or the cultivation of plants on a medium containing the auxin inhibitor NPA leads to the formation of simplified leaves with smooth leaf margins, reduced serration, and

a smaller number of leaflets. This simplification is due to the elimination of the discrete distribution of auxin reactions at the leaf edges, which is necessary for the formation of distinct and separate leaflets [42]. In this study, the results showed that the auxin levels were significantly higher in *Pts*, were nonsignificantly lower in *tf2*, and were significantly lower in *e* than that in the wild types (Figure 5A). The same gradual decrease in auxin was consistent with a decrease in the number of compound leaves and leaf complexity. Plants produce active IAA via de novo synthesis and the hydrolysis of IAA conjugates [50,51]. Auxin amino acid hydrolase can hydrolyze auxin conjugates and release free IAA. *IAR3*, which is a member of the auxin amide hydrolase family, is highly expressed in the roots, stems, and flowers [52]. We verified the changes in the expression of auxin amino acid hydrolase in the leaf primordia at different developmental stages using RT-qPCR (Figure 6D). Compared to the control, the *pTRV2-SlIAA-aah* plants showed a short stature and deepened leaf splits (Figures 8D and S3D). Tomato plants with an *IAR3* knockout exhibited lower levels of free IAA and changes in the response to the pathogen challenge. Changes in *IAR3* expression levels led to changes in auxin homeostasis, which ultimately affected the plant defense responses [53]. Further verification is required for whether the differences in leaf morphology and plant height presented by *pTRV2-SlIAA-aah* were caused by the changes in endogenous auxin homeostasis.

GAs are involved in several developmental processes, such as seed germination, stem elongation, trichome development, pollen maturation, and flowering induction [43]. During leaf development, GA regulates cell proliferation and expansion and affects leaf complexity. Increased GA levels or responses in tomatoes result in taller plants; faster leaf maturation; and, therefore, simpler leaves than those in the wild types [24,25,27]. Compared with the wild types, the mutants *Pts*, *tf2*, and *e* showed significant differences in their leaf hair number, cell size, and cell number (Figure 2A–T). Therefore, we speculated that the difference was due to the changes in the GA levels. This conjecture is also consistent with the expression of the GA2ox7 and GA20ox genes that we screened. The exogenous application of $GA_3$ result in a reduced leaf number and smoothened leaf margins [42]. Reduced GA levels or responses result in stunted plants and, in some cases, the development of highly complex leaves [28,54]. The level of $GA_3$ in *Pts* and *tf2* was higher than that in the wild types, while the level of $GA_3$ in *e* was significantly lower than that in the wild type (Figure 5B). Although the trend of the GA level was not consistent with the complexity of leaves, the analysis of the GA biosynthesis pathway explained this phenomenon (Figure 5D). The enzymes that are independently encoded by the *GA2ox*, *GA3ox*, and *GA20ox* genes control GA biosynthesis and catabolism. The expression of *SlGA20 oxidase 1* (*SlGA20ox1*), a GA biosynthetic gene, increased in the *LA* mutants and decreased in the miR319-overexpressing plants. Conversely, the transcript levels of the GA-inactivated gene, *SlGA2 oxidase 4* (*SlGA2ox4*), were increased in the miR319-overexpressing plants. *SlGA2ox4* is expressed in the initial leaflet during early leaf development, and its expression increases the complexity of tomato leaves [54]. *GA2ox7* and *GA20ox* are the GA degradation and synthesis genes, respectively. We used the P3 stage of leaf primordium development as a control, and the results of the RT-qPCR analysis showed that the expression level of *Ga20ox* was relatively high (Figure 6C), while the expression of *Ga2ox7* was significantly increased at the P10 stage (Figure 6B). This result also confirmed that *GA2ox7* and *GA20ox* play roles in different stages of leaf development. The *pTRV2-SlGA2ox7* and *pTRV2-SlGA20ox* plants also showed opposite phenotypes in terms of plant height (Figure S3B,C). Compared with the control, the *pTRV2-SlGA2ox7* plants were taller (Figure S3B); however, there was no significant difference in leaf morphology (Figure 8B). Recent studies have also shown that water deprivation induces the expression of *GA20ox7* in guard cells and leaf tissues, leading to a decrease in the level of active GA, and that a loss in *GA2ox7* attenuates the stomatal response to water deficiencies [55]. Therefore, we believe that *GA2ox7* may mainly act on guard cells, and its transient silencing led to changes in the GA level and the appearance of higher plant phenotypes; however, there was no significant difference in the leaves. The *pTRV2-SlGA20ox* plants were shorter and

had smaller leaves than the control plants and (Figures 8C and S3C). The differences in the plant height and leaf size of *pTRV2-SlGA20ox* were also consistent with the phenotype of plants with decreased GA levels. Therefore, it was further confirmed that the homeostasis of the auxin and gibberellin levels was important for leaf development.

*3.3. Key Regulators of Tomato Compound Leaf Development*

The expression of the *bHLH079*, *WRKY76*, and *WRKY44* transcription factors was gradually decreased in the comparison groups (Tables S6 and S8). The expression of *bHLH079* increased during P4–P6 and significantly decreased during P7–P10 (Figure 6E). Research has shown that the bHLH transcription factor regulates cell elongation in cotton and Arabidopsis, with Arabidopsis leaves and cotton fibers being longer after its overexpression [16,56]. The bHLH gene, *SlPRE5*, is significantly expressed in young leaves, sepals, and flowers [57]. Therefore, we concluded that *bHLH079* was related to leaf development. *IBH1* (an IL1-binding bHLH) overexpression causes erect leaves in rice and dwarfism in Arabidopsis [58], similar to the dwarfing phenotype in the *pTRV2-SlbHLH079* plants (Figure S3E). However, the leaf morphology of the *pTRV2-SlbHLH079* plants was also different. The leaves were smaller, the leaf margins were smooth, and the number of leaflets was reduced (Figure 8E). As described above, leaf complexity (leaf margin type, serration, and number of leaflets) is closely related to the auxin and GA levels. It has also been shown that the bHLH transcription factors can function by regulating hormones [59–62]. The bHLH transcription factors, phytochrome-interacting factors (PIFs), act as important hubs for the light, auxin, GA, and brassinosteroid pathways. PIFs participate in the regulation of leaf senescence and chlorophyll biosynthesis [63]. During the development of pistil, HEC1 regulates the synthesis, transport, and response of auxin to control the auxin and cytokinin levels in *A. thaliana* [64]. Moreover, the GA signal suppressor, the DELLA protein, can interact with the bHLH transcription factors, such as *AtPIFs*, *AtMYC2*, and *AtARFs*, in *A. thaliana* [65–67].

*WRKYs*, another major transcription factor family, are important for plant growth, development, and the responses to abiotic stress. *NtWRKY4* has been shown to be involved in leaf morphogenesis [44]. The expression of *WRKY44* first increased and then decreased at different developmental stages of the leaf primordia (Figure 6F). However, the expression of *WRKY76* was low during the whole period of leaf primordium development (Figure 6G). Although the expression trends of *WRKY44* and *WRKY76* were not consistent at different leaf primordium stages, the differential phenotypes of the *pTRV2-SlWRKY44* and *pTRV2-SlWRKY76* plants were consistent. The leaves of *pTRV2-SlWRKY44* and *pTRV2-SlWRKY76* were etiolated, with smooth leaf margins and no obvious petioles in some parts (Figure 8F,G), which was similar to the phenotype of the *tf2* mutant. In addition, WRKY, as a key transcription factor in the regulation of leaf senescence, plays a role in the regulation of the chlorophyll- and senescence-related genes [68–70] and in the process of leaf senescence through the hormonal pathways [71–74]. These results also indicate that *bHLH079*, *WRKY44*, and *WRKY76* are key factors in leaf development.

In this study, three groups of mutants with different leaf morphologies, *pts*, *tf2*, and *e*, were used to explore the key factors affecting leaf morphology. The differences between the mutants and the wild types were not only reflected in leaf morphology but also in some physiological indexes. Through the screening of the three groups, we guessed that the differences in the transpiration rate, stomatal conductance, cell permeability, and other physiological indicators were closely related to the GDSLs gene. The difference in the endogenous auxin and gibberellin levels also indicated that the change in leaf morphology was closely related to the change in hormone levels. Through further screening and VIGS verification, we also confirmed that the transcription factors bHLH079, WRKY44, and WRKY76 and the hormone-related genes *IAA-aah*, *GA2ox7*, and *GA20ox* were factors affecting leaf morphology.

## 4. Materials and Methods

### 4.1. Plant Materials and Culture Conditions

In this study, VF36, *Petroselinum* (*Pts*), Condine Red (CR), *Trifoliate* (*tf*2), Ailsa Craig (AC), and *Entire* (*e*) were used as experimental materials. Tomato mutants were *Pts*, *tf*2 (*Solanum lycopersicum* cv. CR), and *e* (*Solanum lycopersicum* cv. Ailsa Craig). *Pts* showed multiple leaflets due to the retardation of the leaf development process by the *KNOX1* gene [13]; *tf*2 presented a three-leaflet state due to the promotion of leaf maturation [75]; and *e* showed the functional loss of auxin inhibitory factor *IAA9*, resulting in single leaves formed via leaf primordium fusion [1]. Seeds were maintained in a water bath (55 °C) for 15 min and were then incubated at a constant temperature (37 °C) until germination (3–4 d). All the plants were grown in a controlled chamber (25 °C, 16/8 h light/dark illumination, and 70–75% humidity). The leaves were collected 45 d after germination, which was when they were fully expanded.

### 4.2. RNA Isolation

The fifth mature leaf sample collected at 45 d was immediately frozen in liquid nitrogen and stored at −80 °C until use. Total RNA was isolated using TRIzol Reagent (Invitrogen Life Technologies, Carlsbad, CA, USA), and the concentration, quality, and integrity were determined using a NanoDrop spectrophotometer (Thermo Scientific, Waltham, MA, USA). RNA (3 mg) was used for RNA sample preparation.

### 4.3. Library Preparation

Sequencing libraries were generated using a TruSeq RNA Sample Preparation Kit (Illumina, San Diego, CA, USA). Briefly, mRNA was purified from the total RNA using poly-T oligo-attached magnetic beads. Fragmentation was performed using divalent cations at elevated temperatures in an Illumina proprietary fragmentation buffer. First-strand cDNA synthesis was performed using random oligonucleotides and SuperScript II (Invitrogen, USA). Subsequently, second-strand cDNA synthesis was performed using DNA Polymerase I and RNase H. Remaining overhangs were converted into blunt ends via exonuclease/polymerase activities, and the enzymes were removed. After adenylation of the 3′ ends of the DNA fragments, Illumina PE adapter oligonucleotides were ligated to prepare them for hybridization. To select the preferred 200 bp cDNA fragments, library fragments were purified using the AMPure XP system (Beckman Coulter, Beverly, CA, USA). DNA fragments with ligated adaptor molecules at both ends were selectively enriched using the Illumina PCR Primer Cocktail in a 15-cycle PCR reaction. The products were purified (AMPure XP system) and quantified using an Agilent high-sensitivity DNA assay on a Bioanalyzer 2100 system (Agilent, California, USA). The sequencing library was sequenced on a HiSeq platform (Illumina) by Shanghai Personal Biotechnology Co. Ltd.

### 4.4. Sequencing and Data Analysis

Eighteen RNA libraries were constructed from *Pts*, *tf*2, *e*, and their respective wild-type leaf samples. Raw reads were generated via high-throughput sequencing of the six samples with three replicates each (Table S1). Clean reads were obtained by removing the adapters and low-quality reads from the raw data. Using HISAT2 2.1.0 software, reads were filtered to *Solanum lycopersicum* genome [76]. HTSeq 0.9.1 was used to compare the read count value for each gene, and FPKM was used to standardize the expression level of the original gene. For the DESeq variance analysis, DEGs were screened based on the following conditions: multiple expression differences, log2FoldChange > 1, and significant *p*-value < 0.05. A BLAST search was performed on sequences of DEGs against the database in NCBI. The GO enrichment analysis was performed by GO (http://geneontology.org/, on 2 December 2022), and the KEGG enrichment analysis was performed by Kyoto Encyclopedia of Genes and Genomes (http://www.kegg.jp/, on 2 December 2022).

### 4.5. Reverse Transcription Quantitative PCR (RT-qPCR) Analysis

RNA extraction and real time-qPCR were performed using 45 d old leaves and the RNAiso plus reagent (TaKaRa, Shiga, Japan). First-strand cDNA was synthesized using a HiScript® II 1st Strand cDNA Synthesis Kit (Vazyme, Nanjing, China). RT-qPCR was performed using AceQ®qPCR SYBR Green Master Mix (Vazyme, Nanjing, China); the primers are listed in Supplementary Table S1. The $2^{-\Delta\Delta Ct}$ method was used to analyze relative gene expression. Data were reported as the mean of three replicates. The housekeeping gene β-actin (primers: 5′-TGTCCCTATTTACGAGGGTTATGC-3′ (actin-F) and 5′-AGTTAAATCACGACCAGCAAGAT-3′ (actin-R)) was used as the internal control. The PCR program was as follows: 95 °C for 45 s, 40 cycles of 95 °C for 10 s, 58 °C for 25 s, and 72 °C for 20 s. For all RT-qPCR experiments, at least three biological replicates were performed, and each reaction was run in triplicate.

### 4.6. Scanning Electron Microscopy

For scanning electron microscope images, 45 d old fifth leaves were fixed in 2.5% glutaraldehyde in 0.1 mol phosphate buffer (pH 7.4), were dehydrated in a graded ethanol series, were attached with colloidal graphite to a copper stub, and were frozen under vacuum. Dried samples sputtered with 25–30 nm of gold palladium (SC-500; Soquelec, Montreal, QC, Canada) were imaged using a SEM (JSM-6390LV; NTC, Tokyo, Japan) at an accelerating voltage of 10 kV.

### 4.7. Transpiration Rate Measurements

The fifth mature leaf at 45 d was used to measure the transpiration rate with a portable photosynthesis system (LI-6400XT; Li-Cor, Lincoln, NE, USA) under 150 μmol m$^{-2}$ s$^{-1}$ light intensity, 70–75% relative humidity, and 430 ppm $CO_2$ concentration. The measurements were recorded every 30 s for 30 min. Data were presented as the means of at least three leaves from individual plants for each experiment. The experiments were repeated at least three times.

### 4.8. Tissue Collection

L5 P1 is the fifth leaf when it has just initiated from the SAM, and it becomes L5 P2 after the initiation of the next primordium. For very young leaf primordia at the P1–P3 stages, the leaves were collected with younger leaf primordia and SAM. At the P4 and P5 stages, leaves were collected both with and without the SAM and younger primordia [15].

### 4.9. Statistical Analysis

One-tailed *t*-test was used to compare the significance of differences in gene expression data from the RT-PCR analysis. One-way ANOVA was used to compare the contents of DEGs at P3–P10 developmental stages. All values were reported as mean ± standard error (SE). Differences were considered statistically significant at $p < 0.05$. All data analyses were performed using SPSS 19.0 (SPSS Inc., Chicago, IL, USA).

### 4.10. UFLC-ESI-MS/MS Assay

The endogenous IAA and $GA_3$ contents were determined using UFLC-ESI-MS/MS. Mature leaves were collected and ground using liquid nitrogen. Three replicates were prepared for each leaf sample. The biomass for each replicate was 0.5 g. Subsequently, IAA and $GA_3$ contents were determined according to protocol published by ESI-MS/MS [77].

### 4.11. VIGS Assays

For the construction of the VIGS vector, 300 bp fragments of the genes *SlbHLH079*, *SlGA2ox7*, *SlGA20ox*, *SlIAA-amino acid hydrolase*, *SlWRKY44*, and *SlWRKY76* were amplified using sequence-specific primers (Supplementary Table S2). The pTRV1-, pTRV2-, and pTRV2-host target genes were transformed into *Agrobacterium tumefaciens* strain GV3101 via electroporation. Cultures containing pTRV1 and pTRV2 vectors were mixed in a 1:1 ratio

either individually or simultaneously. The prepared liquid was used to infect the AC cotyledons. The infected plants were transferred to a culture chamber. The infected plants were then transferred to a growth chamber (16 h day/8 h night cycle, 25 °C, and 70–75% humidity). The phenotypes were analyzed 6–7 weeks after inoculation.

*4.12. Chlorophyll Leaching Assay*

Leaf epidermal penetration of 45 d old plants was measured. First, 0.2 g of leaves was cut into uniform cubes and was soaked in 20 mL of 95% anhydrous ethanol. The absorbance of the samples was measured every 20 min. The treated samples were placed in the dark for 24 h to determine the total chlorophyll content. An ultraviolet (UV) U-5100 spectrophotometer was used to measure the absorbance of the samples at wavelengths of 663 and 645 nm. Total micromoles of chlorophyll were calculated as 20.21 (A645) + 8.02 (A663).

$$\text{Chlorophyll content} = \text{concentration} \times \text{total extracted liquid (mL)/sample fresh weight} \times 1000.$$

**Supplementary Materials:** The following supporting information can be downloaded at https://www.mdpi.com/article/10.3390/horticulturae9030363/s1, Table S1. The profiles of RNA deep sequencing for VF36, *Pts*, CR, *tf2*, AC, and *entire*; Table S2. Primers used in this study; Table S3. RNA-Seq analysis of DEGs in *Pts*/VF36; Table S4. RNA-Seq analysis of DEGs in *tf2*/CR; Table S5. RNA-Seq analysis of DEGs in *entire*/AC; Table S6. RNA-Seq analysis of DEGs in *Pts*/VF36 and *tf2*/CR; Table S7. RNA-Seq analysis of DEGs in *Pts*/VF36 and *entire*/AC; Table S8. RNA-Seq analysis of DEGs in *tf2*/CR and *entire*/AC; Figure S1. Morphological differences between mutant and wild-type plants; Figure S2. GO and KEGG enrichment analyses; and Figure S3. Differences in leaf morphology of VIGS plants compared with that of pTRV2 plants.

**Author Contributions:** G.Z., X.L. and J.J. conceived this project and designed the work. G.Z., C.M., S.Y. and X.Z. performed the research, analyzed the data, and wrote the manuscript. G.Z., X.L. and J.J. revised the paper. All authors have read and agreed to the published version of the manuscript.

**Funding:** This work was supported by the National Key Research and Development Program of China (2019YFD1000301) and the National Natural Science Foundation of China (32172548, 31801847).

**Data Availability Statement:** Data Availability Statement: All data generated or analyzed during this study are included in this published article (and supplementary information files).

**Acknowledgments:** In this study, VF36, Petroselinum (Pts), Condine Red (CR), Trifoliate (tf2), Ailsa Craig (AC), and Entire (e), which were used as experimental materials, were from the Tomato Genetics Resource Center at the University of California, Davis. We hereby express our thanks.

**Conflicts of Interest:** The authors declare that the research was conducted in the absence of any commercial or financial relationships that could be construed as a potential conflict of interest.

## Abbreviations

| | |
|---|---|
| ABA | Abscisic acid |
| AC | Ailsa Craig |
| ARF | Auxin response factor |
| bHLH | Basic helix-loop-helix |
| BLAST | Basic local alignment search tool |
| BR | Brassinosteroids |
| CKs | Cytokinins |
| CKX | Cytokinin oxidase genes |
| CR | Condine Red |
| CPS | Ent-copalyl diphosphate synthase |
| DEGs | Differentially expressed genes |
| dpy | Defective mutant dumpy |
| *e* | Entire |
| ET | Ethylene |

| | |
|---|---|
| FPKM | Fragments per kilobase of exon model per million mapped reads |
| GA | Gibberellic acid |
| GA2ox7 | Gibberellin 2-beta-dioxygenase 7 |
| GA20ox | Gibberellin 20-oxidase-like |
| GDSL | GDSL esterases/lipases |
| GO | Gene ontology |
| GhFP1 | Cotton bHLH protein |
| HD | Homeodomain |
| IAA | Indole-3-acetic acid |
| IAA-aah | IAA-amino acid hydrolase |
| IBH1 | ILI1-binding BHLH Protein 1 |
| IPT7 | Iso-pentenyl transferase 7 |
| JA | Jasmonic acid |
| KAO | Ent-kaurenoic acid oxidase |
| KEGG | Kyoto Encyclopedia of Genes and Genomes |
| KNOX | Knotted1-like homeobox |
| KO | Ent-kaurene oxidase |
| KS | Ent-kaurene synthase |
| LA | LANCEOLATE |
| *LP1* | Leaf-associated protein 1 |
| *LP2* | Leaf-associated protein 2 |
| MYC | Myelocytomatosis |
| OCL | Outer cuticular ledge |
| OSP1 | Occlusion of Stomatal Pore 1 |
| PIFs | Phytochrome-interacting factors |
| PM | Primary morphogenesis |
| Pts | Petroselinum |
| SAGs | Senescence-related gene |
| SAM | Shoot apical meristem |
| SM | Secondary morphogenesis |
| *tf*2 | Trifoliate |
| TFs | Transcription factors |
| VIGS | Virus-induced gene silencing |
| *wdl1* | Wilted dwarf and lethal 1 |
| WOX | WUSCHEL-related homeobox |

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
