# Peer review of "Transcriptome Analyses Reveal the Key Regulators of Tomato Compound Leaf Development"

_horticulturae, doi:10.3390/horticulturae9030363_

Round 1
Reviewer 1 Report
The objective is not described, I suggest expanding the justification because it is not clear
I suggest that the title concepts are not included in the keywords
![]()

Figure 1. Morphological differences between mutant and wild-type plants. (A) Pts mutant had sig-131 nificantly greater number of leaflets than VF36 wild-type. (B) tf2 mutant had significantly lesser 132 number of leaflets and deeper cleft than CR wild type. (C) e mutant had simplified leaves and 133 smooth leaf margins compared with AC wild type. (A–C) 45-d-old plants. The selected leaf was the 134 fifth true leaf of the 45-d-old plant in (A–C). Bars = 500 μm in (A–C).
It is necessary to point out the structures mentioned in the description of the image (point structure names inside image), moreover, the images must be mentioned in the order in which they are arranged. (VF36 then PTS, CR then tF2… etc)
Figure 2. Differences in the number of leaf hair and cells between tomato mutant and wild type 156 visualized using scanning electron microscope (SEM). (A–F) Images of mutant Pts, tf2, e, and wild-157 type VF36, CR, and AC under SEM. Bars = 50 μm in (A–F). (G) Differences in the number of leaf 158 hair between mutant Pts, tf2, e, and wild-type VF36, CR, and AC. Values are presented as means ± 159 SE (n = 3). * and ** indicate significant differences between wild type and transgenic lines or mutants 160 at P < 0.05 and P < 0.01, respectively. (H–S) Images of epidermal cells observed under SEM at 300 161 (H, J, L, N, P, R) and 800 (I, K, M, O, Q, S) magnification. Bars = 50 μm in (H, J, L, N, P, R). Bars = 162 50 μm in (I, K, M, O, Q, S). (T) The number of epidermal cells differed between mutant Pts, tf2, entire, 163 and wild-type VF36, CR, and AC. Values are presented as means ± SE (n = 3). * and ** indicate 164 significant differences between wild type and transgenic lines or mutants at P < 0.05 and P < 0.01, 165 respectively. Redaction is not clear, better summarize and prioritize information, or rather select the most important out of all the images for better understanding and avoid confusing extra information.
Transpiration rate, stomatal conductance, and epidermal cell permeability are closely related 225 to GDSLs expression avoid abbreviations in the titles of each section.
The work is extensive, I suggest reporting the relevant results and carefully selecting the images to be published. The way it is presented results in boring work and is not clear.
The results may be interesting but there is no clear discussion. What is the importance of this research?
The discussion is repetitive and no conclusions were presented.

Author Response
Dear reviewer:
We are truly grateful to the reviewers’ suggestions. Based on these comments, we have made careful modifications on the original manuscript. Below you will find our point-by-point responses to the reviewers’ comments/questions.
Response to reviewers
Thank reviewers for constructive suggestions and comments. Our responses are below:
Major Point:
- The objective is not described, I suggest expanding the justification because it is not clear.
We have followed your suggestions and in view of this problem, we have made the goal clear in the Abstract of the new manuscript in Line in red.
- I suggest that the title concepts are not included in the keywords.
We have followed your suggestions and deleted key words from transcriptome analysis in Line26 in the new manuscript.
- It is necessary to point out the structures mentioned in the description of the image (point structure names inside image), moreover, the images must be mentioned in the order in which they are arranged. (VF36 then Pts, CR then tF2… etc).
We have followed your suggestions and the leaf structure picture (D) is added in Figure 1 in Line131, and the leaf structure information is explained in the picture annotation in Line136 in the New Manuscript in red.
- Redaction is not clear, better summarize and prioritize information, or rather select the most important out of all the images for better understanding and avoid confusing extra information.
We have followed your suggestions and in response to your note on Figure 2, we have deleted and summarized the duplicate information in the new manuscript to facilitate reading in Line161-168.
- Transpiration rate, stomatal conductance, and epidermal cell permeability are closely related 225 to GDSLs expression avoid abbreviations in the titles of each section.
We have followed your suggestions and the full name of the gene was modified in the new manuscript in Line 228 in red.
- The work is extensive, I suggest reporting the relevant results and carefully selecting the images to be published. The way it is presented results in boring work and is not clear.
We have followed your suggestions and we have deleted some pictures and captions in the new manuscript in Line 232-253, 261-280, 406-410 in red.
- The results may be interesting but there is no clear discussion. What is the importance of this research? The discussion is repetitive and no conclusions were presented
Leaf morphology affects plant stress resistance, transpiration, photosynthesis and other physiological indicators, as well as fruit quality and yield. Therefore, it is very important to explore the key factors affecting leaf morphology. In this study, three groups of mutants with different leaf morphology were used to further explore the key factors affecting leaf morphology. The three groups of mutants not only had differences in leaf morphology, but also had differences in some physiological indexes. The differences in transpiration rate, stomatal conductance and epidermal cell permeability may be closely related to the differentially expressed gene GDSLs. Moreover, the endogenous auxin and gibberellin levels were also different, so the change of leaf morphology of the mutant was also related to the change of this hormone level. Finally, VIGS were used to verify candidate genes bHLH079, WRKY44, WRKY76, IAA-aah, GA2ox7, and GA20ox, confirming that these six genes are factors affecting leaf morphology.
We provide a brief summary of the results at the end of the discussion of the new manuscript in lines 521-530, in red.

Reviewer 2 Report
I have two comments that I think of very important to address by the authors:
1. A literature review is a summary of what research has been completed in a topic area; it should be summarized in your own words.
2. Authors need to improve the quality of the figures.
Author Response
Dear reviewer:
We are truly grateful to the reviewers’ suggestions. Based on these comments, we have made careful modifications on the original manuscript. Below you will find our point-by-point responses to the reviewers’ comments/questions.
Response to reviewers
Thank reviewers for constructive suggestions and comments. Our responses are below:
- A literature review is a summary of what research has been completed in a topic area; it should be summarized in your own words.
We have followed your suggestions and have revised them in the new manuscript.
- Authors need to improve the quality of the figures.
The resolution of our picture is 300bpi, which meets the requirements. As for the photo problem you mentioned, the resolution is automatically reduced when you insert it into word. We can provide you with the original images as a zip file.

Reviewer 3 Report
The manuscript deals with transcriptome analyses to reveal the key regulators of tomato compound leaf development. Three transcription factors (bHLH079, WRKY44, and WRKY76) and three hormone-regulated genes (IAA-amino acid hydrolase, Gibberellin2ox7, and Gibberellin20ox) that were differentially expressed in the transcriptome were subjected to detailed analysis. The study is soundly performed and well-written. The authors use not only the comparative analysis of the transcriptomes of tomato mutants with varying leaf structures but also apply virus-induced gene silencing to confirm the roles of the candidate genes identified. That said I have the following comments/suggestions :
1. The last part of the Introduction (Lines 101-116) should be moved to the Materials and methods section.
2. One sentence (Lines 168-170) seems to be a result of merging at least two former sentences. It should be rewritten for clarity.
3. Panel O of Figure 5 represents only chlorophyll content at 24 h of the entire mutant as significantly different from the wild-type (VF36) plants. This seems incorrect as the content of this mutant (as displayed in the figure) is exactly the same as in the VF36 accession, while the other mutants seem to have much higher content (which is not shown as significantly different according to the description provided in the caption). This needs to be better described/explained.
Author Response
Dear reviewer:
We are truly grateful to the reviewers’ suggestions. Based on these comments, we have made careful modifications on the original manuscript. Below you will find our point-by-point responses to the reviewers’ comments/questions.
Response to reviewers
Thank reviewers for constructive suggestions and comments. Our responses are below:
- The last part of the Introduction (Lines 101-116) should be moved to the Materials and methods section.
We have followed your suggestions and move this sentence to the Materials method in the new manuscript in Line531-535 in red.
- One sentence (Lines 168-170) seems to be a result of merging at least two former sentences. It should be rewritten for clarity.
- Panel O of Figure 5 represents only chlorophyll content at 24 h of the entire mutant as significantly different from the wild-type (VF36) plants. This seems incorrect as the content of this mutant (as displayed in the figure) is exactly the same as in the VF36 accession, while the other mutants seem to have much higher content (which is not shown as significantly different according to the description provided in the caption). This needs to be better described/explained.
First, we used three sets of mutants, Pts/VF36, tf2/CR, and entire/ AC. Therefore, the data were compared by comparing the mutant with the corresponding wild type. The cell permeability results showed that the leaching rate of the entire mutant was significantly lower than that of the wild-type AC at the later time. To account for this, we measured the total chlorophyll content, which showed that the entire mutant was significantly lower than the wild-type AC. Thus, it was explained why the leaching rate of the entire mutant was lower than that of the wild-type AC at the later stage.

Round 2
Reviewer 1 Report
No comments